# Intrinsic and Extrinsic Transcriptional Profiles That Affect the Clinical Response to PD-1 Inhibitors in Patients with Non–Small Cell Lung Cancer

**DOI:** 10.3390/cancers15010197

**Published:** 2022-12-29

**Authors:** Hye Eun Byeon, Seokjin Haam, Jae Ho Han, Hyun Woo Lee, Young Wha Koh

**Affiliations:** 1Institute of Medical Science, Ajou University School of Medicine, Suwon 16499, Republic of Korea; 2Department of Thoracic and Cardiovascular Surgery, Ajou University School of Medicine, Suwon 16499, Republic of Korea; 3Department of Pathology, Ajou University School of Medicine, Suwon 16499, Republic of Korea; 4Department of Hematology-Oncology, Ajou University School of Medicine, Suwon 16499, Republic of Korea

**Keywords:** non-small cell lung cancer, PD-L1, PD-1, PD-1/PD-L1-targeted therapy, biomarkers, immunotherapy, tumor-intrinsic role, gene expression profile, machine learning

## Abstract

**Simple Summary:**

Monoclonal antibodies targeting the programmed death 1 (PD-1) receptor and its ligand (PD-L1) have demonstrated improved clinical response and survival in non-small cell lung cancer (NSCLC). Although extrinsic immunologic factors play important roles in the regulation of PD-L1 and PD-1, tumor intrinsic factors, including genetic alterations, epigenetic alterations, oncogenic and tumor suppressor signals, and transcription factors also play important roles in PD-L1 expression. There is an urgent need to investigate the intrinsic transcriptional profiles affecting the clinical response to PD-1 inhibitors in patients with non-small cell lung cancer. In our study, PD-1 inhibitor-associated intrinsic gene patterns were very different between lung adenocarcinoma and squamous cell carcinoma. In lung adenocarcinoma, the intrinsic gene signature was a very good predictive or prognostic biomarker. Our findings prove for the first time that an intrinsic gene signature is well predictive of responsiveness to PD-1 inhibitors in lung adenocarcinoma.

**Abstract:**

Using a machine learning method, we investigated the intrinsic and extrinsic transcriptional profiles that affect the clinical response to PD-1 inhibitors in 57 patients with non-small cell lung cancer (NSCLC). Among the top 100 genes associated with the responsiveness to PD-1 inhibitors, the proportion of intrinsic genes in lung adenocarcinoma (LUAD) (69%) was higher than in NSCLC overall (36%) and lung squamous cell carcinoma (LUSC) (33%). The intrinsic gene signature of LUAD (mean area under the ROC curve (AUC) = 0.957 and mean accuracy = 0.9) had higher predictive power than either the intrinsic gene signature of NSCLC or LUSC or the extrinsic gene signature of NSCLC, LUAD, or LUSC. The high intrinsic gene signature group had a high overall survival rate in LUAD (*p =* 0.034). When we performed a pathway enrichment analysis, the cell cycle and cellular senescence pathways were related to the upregulation of intrinsic genes in LUAD. The intrinsic signature of LUAD also showed a positive correlation with other immune checkpoint targets, including CD274, LAG3, and PDCD1LG2 (Spearman correlation coefficient > 0.25). PD-1 inhibitor-related intrinsic gene patterns differed significantly between LUAD and LUSC and may be a particularly useful biomarker in LUAD.

## 1. Introduction

Since anti-programmed cell death protein 1 (PD-1) drugs were approved for use in patients with advanced non-small cell lung cancer (NSCLC), a revolutionary change has occurred in NSCLC treatment [1]. Although programmed death ligand 1 (PD-L1) immunohistochemical expression has so far been considered the best option for predicting a patient’s response to PD-1 inhibitors, responses have also been reported in PD-L1-negative patients [2]. Several other predictors, including tumor mutational burden, gut microbial diversity, and mutations in the b-catenin or STK11 gene, have been reported as candidate biomarkers that complement the predictive role of PD-L1, but those biomarkers require further validation [3].

Cancer transcriptional profiles are also attracting attention as biomarkers. Ayers et al. reported that IFN-γ-associated transcriptional profiles predict the clinical response to PD-1 blockade in melanoma, head and neck squamous cell carcinoma (HNSCC), and gastric cancer [4]. Ayers’ IFN-γ-associated genetic signature was also found to be a good predictor of response to PD-1 inhibitors in groups of HNSCC [5] and NSCLC [6] patients by other researchers. It is thought that the extrinsic immune system including IFN-γ, TNF-α, and interleukins plays an important role in the regulation of PD-L1 and PD-1, and therefore, only the transcriptional profile related to immune-related genes has been mainly studied. However, it has recently been shown that tumor-intrinsic signals also play an important role in the expression of PD-L1/PD-1. Several intrinsic signals, meaning non-immune signals, affect PD-L1 expression, including genetic alterations, epigenetic alterations, oncogenic and tumor suppressor signals, transcription factors, and cancer stem cell signals [7]. Although the transcriptional profile of such intrinsic signals could influence resistance or relapse in response to PD-1 inhibitor treatment, it has not been studied at all. Lung adenocarcinoma (LUAD) and lung squamous cell carcinoma (LUSC) have very different mutation profiles that are known to significantly affect treatment and prognosis [8]. Therefore, intrinsic signals should be analyzed by cell subtype because different intrinsic mutation profiles affect PD-L1/PD-1 expression differently in different cell subtypes.

Machine learning approaches are now being widely used for biomarker discovery [9]. Since the application of that technology, the accuracy of predicting cancer prognosis has improved by 15–20% [10]. Two important components of machine learning are classification and feature selection. In this study, we used six classification models, and the feature selection method was the analysis of variance (ANOVA).

We analyzed the transcriptional profiles of 57 NSCLC patients who received PD-1 inhibitor monotherapy and classified 780 genes as intrinsic or extrinsic according to their mechanism. Machine learning was used to determine whether a transcriptional profile could predict a patient’s responsiveness to PD-1 inhibitor treatment. A subgroup analysis was performed according to the cellular subtype. A pathway enrichment analysis was performed to analyze the major pathways of the selected genes.

## 2. Patients and Methods

### 2.1. Patients and PD-L1 Test

This study was approved by the Institutional Review Board of the Ajou University School of Medicine (AJIRB-BMR-KSP-20-396). Informed consent was waived due to the retrospective study design. We retrospectively selected 57 patients with advanced NSCLC who received PD-1 inhibitors from 2016 to 2021. The patients’ clinicopathological characteristics are summarized in Appendix A. All patients underwent lung biopsy or surgical resection. Inoperable cases were treated with PD-1 inhibitors, and surgical cases were treated with PD-1 inhibitors for subsequent recurrence or metastasis. Among the 57 patients, 31 had LUAD and 26 had LUSC. Patients treated with PD-1 inhibitors were classified as responders (complete response or partial response) or non-responders (stable disease or disease progression) [11]. The PD-L1 immunohistochemistry (clone: SP263, rabbit monoclonal, Roche, Basel, Switzerland) was performed using the OptiView DAB Immunohistochemical Detection Kit on a Ventana BenchMark ULTRA instrument.

### 2.2. Gene Expression Analysis

We used the nCounter^®^ Tumor Signaling 360 Panel to examine both intrinsic and extrinsic genes related to a response to PD-1 inhibition [12]. The nCounter^®^ Tumor Signaling 360 panel contains 760 genes considered important for tumor biology. We added 20 genes related to PD-L1 expression and thus investigated a total of 780 genes. The selection method for the additional genes is described in Appendix A. The NanoString company divided the 760 genes in the panel into 10 pathways according to the criteria of Hanahan et al. [13]. The pathways of the additional 20 genes were also divided according to the same criteria. The pathways of all 780 genes are summarized in Appendix A. Of the 10 pathways, the avoiding-immune-destruction and tumor-promoting-inflammation pathways were defined as extrinsic pathways, and the remaining 8 pathways were defined as intrinsic pathways. Because our interest is tumor-intrinsic pathways, we marked and extracted only intratumoral regions from the formalin-fixed paraffin-embedded (FFPE) blocks. After RNA extraction from the FFPE tissue, reporter code and capture probe sets were mixed, and the hybridization reaction was performed, followed by a high-sensitivity procedure at the preparation station. Sample scans were performed using an nCounter Digital Analyzer, and then normalization was performed using the geometric mean of the positive control counts and a housekeeping gene.

### 2.3. Machine Learning Approach and Statistical Analysis

We performed a machine learning analysis using mRNA data obtained from the NanoString analysis. The feature reduction method is a process for selecting genes that can predict the response to a PD-1 inhibitor from our pool of 780 genes. We used ANOVA, which is often used in gene expression analyses, as the feature reduction method [14,15,16]. Next, it was necessary to determine the number of genes needed to predict a patient’s response to a PD-1 inhibitor, and to do that, the machine learning algorithm selected the number of genes with the highest area under the receiver operating characteristic curve (AUC) value. The AUC value of each gene set was derived using six machine learning classifiers: the naïve Bayes method (NB), neural network (NN), random forest (RF), logistic regression (LR), support vector machine (SVM), and k-nearest neighbor (kNN). In this case, the AUC value indicates how well a set of genes predicts a patient’s response to PD-1 inhibitors. To evaluate the performance of the predictive classification models, we used the leave-one-out cross-validation (LOOCV) procedure because our dataset was relatively small. In LOOCV, the number of folds is set to the number of examples in the dataset. Therefore, the learning algorithm is applied once for each example, using all other examples as a training set and the selected example as a test set. In other words, if the number of samples is n, the entire procedure is repeated n times. We used five indicators to measure the performance of each machine learning model: AUC, accuracy (the rate of correct classification), F1 score (the harmonic mean of the model’s precision and recall), precision (positive predictive value), and recall (sensitivity). After calculating the sum of those five performance scales for each machine learning model, we chose the model with the highest sum as the most suitable. We then performed a survival analysis using the value predicted by the most suitable model.

### 2.4. Statistical Analysis

Fold changes, normalized data, and *p*-values were obtained through nSolver, a NanoString analysis tool. For the statistical analysis, we used IBM SPSS Statistics 25 software (IBM, Armonk, NY, USA) or R version 3.5.3 (The R Foundation), and statistical significance was set at *p*-value < 0.05. We used Orange version 3.27 software (Bioinformatics Laboratory at the University of Ljubljana, Slovenia) for machine learning approach [17]. We performed the Gene Set Enrichment Analysis (GSEA) using GSEA version 4.0.3. http://www.broadinstitute.org/gsea/index.jsp (accessed on 7 April 2022) [18]. The DAVID Bioinformatics Resources 6.8 tool [19] was used for the Kyoto Encyclopedia of Genes and Genomes (KEGG) pathway analysis.

## 3. Results

### 3.1. Feature Selection and Prediction Modeling Using Machine Learning Methods

We sequentially selected 100 genes involved in the response to PD-1 inhibitors using ANOVA. Among those 100 genes, the extrinsic and intrinsic pathways were selected separately according to the annotation in the NanoString guideline. Among our 57 NSCLC patients, we found 36 intrinsic pathways and 64 extrinsic pathways (Appendix A). In the LUAD patients, we found 69 intrinsic pathways and 31 extrinsic pathways (Appendix A), and in the LUSC patients, we found 33 intrinsic pathways and 67 extrinsic pathways (Appendix A). To determine the optimal number of genes, we compared the AUCs of the six machine learning methods for between 5 and 30 genes (Appendix A). Among all 57 NSCLC patients, 30 genes had the highest sum of AUCs for intrinsic pathway-related genes, and 5 genes had the highest sum of AUCs for extrinsic pathway-related genes. In LUAD, 15 genes had the highest sum of AUCs for both intrinsic and extrinsic pathway-related genes, and in LUSC, 5 genes had the highest sum of AUCs for intrinsic pathway-related genes, and 10 genes had the highest sum of AUCs for extrinsic pathway-related genes (Appendix A). The gene list for each pathway is summarized in Table 1.

Among all 57 NSCLC patients, the AUC values for the 30 intrinsic pathway-associated genes ranged from 0.71 to 0.87 (NB: 0.858 NN: 0.808, RF: 0.828, LR: 0.719, SVM: 0.879, kNN: 0.743, sum of AUCs: 4.835, Figure 1A). The group predicted by the NN using 30 intrinsic pathway-related genes had a statistically significantly better prognosis than the other patients in the survival analysis (*p =* 0.037, Figure 1B). The AUCs of the 5 extrinsic pathway-associated genes ranged from 0.74 to 0.823 (NB: 0.823 NN: 0.816, RF: 0.807, LR: 0.785, SVM: 0.780, kNN: 0.745, sum of AUCs: 4.756, Figure 1C), and the group predicted by the SVM had the best prognosis (*p =* 0.062, Figure 1D). Figure 1E is an AUC graph of the immunohistochemical expression of PD-L1, and it shows the relatively low value of 0.538; furthermore, PD-L1 expression did not affect survival in the survival analysis (*p =* 0.92, Figure 1F). All performance data from the machine learning are summarized in Appendix A.

In LUAD, 15 intrinsic pathway-associated genes showed AUC values greater than 0.9 with all six classifiers (Figure 2A). In general, if an AUC value is 0.9 or higher, the predictive ability of the related model is considered to be very good, so all 15 intrinsic pathway-related genes in this analysis had very good predictive ability. We confirmed that the group predicted by the LR for intrinsic pathway-related genes had a prognosis that was statistically significantly better than that of other patients in the survival analysis (*p =* 0.034, Figure 2B). However, the 15 extrinsic pathway-related genes had lower AUC values than the intrinsic pathway-related genes (Figure 2C), and the group predicted by the SVM had a better prognosis than other patients, but the difference was not statistically significant (*p =* 0.071, Figure 2D). Figure 2E is an AUC graph of the immunohistochemical expression of PD-L1. It had a relatively low AUC value of 0.67 and did not affect survival in our survival analysis (*p =* 0.91, Figure 2F).

In LUSC, 5 intrinsic pathway-associated genes showed AUCs lower than those in LUAD (NB: 0.9, NN: 0.842, RF: 0.783, LR: 0.742, SVM: 0.808, kNN: 0.708, sum of AUCs: 4.783, Figure 3A). The group predicted by the NN using 5 intrinsic pathway-related genes had a prognosis that was statistically significantly better than that of the other patients in the survival analysis (*p =* 0.0085, Figure 3B). The 10 extrinsic pathway-related genes showed AUC values similar to those of the 5 intrinsic pathway-related genes (NB: 0.883, NN: 0.675, RF: 0.871, LR: 0.525, SVM: 0.792, kNN: 0.675, sum of AUCs: 4.421, Figure 3C), and the group predicted by the SVM had a significantly better prognosis than other patients (*p =* 0.018, Figure 3D). Figure 3E is an AUC graph of the immunohistochemical expression of PD-L1 that shows that the relatively low value of 0.67 and PD-L1 expression did not affect survival in the survival analysis (*p =* 0.98, Figure 3F). All performance data from the machine learning are summarized in Appendix A.

Currently, the most widely known immune-related gene signature correlated with the response to PD-1 inhibitors is the IFN-γ-related mRNA profile published by Ayers et al. [4]. They used 18 expanded immune gene signatures. In this work, we studied 13 genes because 5 of the 18 genes used by Ayers et al. were not included in the NanoString panel. Those 13 extended immune gene signatures from Ayers et al. are summarized in Appendix A. In LUAD, NB had an AUC of 0.714, but the remaining five machine learning models had AUCs < 0.6 (Appendix A). Ayers’ 13 extended immune gene signatures did not correlate with overall survival (OS) in the LR (*p =* 0.43, Appendix A). In LUSC, the LR had the highest AUC of 0.85, and NB, NN, and RF had AUCs > 0.7 (Appendix A). Ayers’ 13 extended immune gene signatures correlated with better prognosis in the LR survival analysis (*p =* 0.0019, Appendix A). All performance data for the machine learning of Ayers’ 13 genes are summarized in Appendix A. Because EGFR mutations can affect the response to PD-1 inhibitors, we performed a subgroup analysis. Since all cases with EGFR mutation were non-response, only the group without EGFR mutation was analyzed. The AUC values for the 15 intrinsic pathway-associated genes ranged from 0.87 to 1 (NB: 1, NN: 1, RF: 0.870, LR: 0.987, SVM: 0.987, kNN: 0.883, sum of AUCs: 5.727, Appendix A). The AUC values for the 15 extrinsic pathway-associated genes ranged from 0.5 to 1 (NB: 0.922 NN: 0.961, RF: 0.805, LR: 0.506, SVM: 1, kNN: 0.786, sum of AUCs: 4.98, Appendix A). Therefore, even in the EGFR-negative group, intrinsic pathway-associated genes showed slightly higher predictive power than extrinsic.

### 3.2. Survival Analysis Using TCGA Dataset

In our dataset, we confirmed that the prognosis was good in the group with a high intrinsic-related gene signature. Although it is best to apply our gene signature to patients treated with PD-1 inhibitors, most past studies have not looked at intrinsic genes or analyzed them by cellular subtype. Therefore, we performed a survival analysis on data from The Cancer Genome Atlas (TCGA) for patients who were not treated with a PD-1 inhibitor. The intrinsic gene signature was calculated as follows: (sum of normalized upregulated gene expression) − (sum of normalized downregulated gene expression). In LUAD, a high intrinsic gene signature correlated with worse survival (*p* < 0.001, Appendix A). In LUSC, a high intrinsic gene signature also correlated with worse survival (*p =* 0.022, Appendix A).

### 3.3. Pathway Analyses

KEGG pathway enrichment analyses were performed for intrinsic and extrinsic pathway-related genes found in the ANOVA. The genes upregulated and downregulated in the responders were analyzed separately. Then, we selected the five pathways with the lowest *p* values. Among the intrinsic genes of LUAD, the cell cycle and cellular senescence pathways are related to the upregulated genes, and the MAPK signaling pathway, focal adhesion, and lysine degradation are related to the downregulated genes (Figure 4A). In LUAD, all 5 pathways associated with the upregulated extrinsic genes, including the cytokine–cytokine receptor interaction, are immune-related pathways (Figure 4B). The intrinsic genes upregulated in LUSC are related to the cellular senescence and HIF-1 signaling pathways, and proteoglycans in cancer and the Hedgehog signaling pathway are related to the downregulated genes (Figure 4C). All 5 pathways related to the extrinsic genes upregulated in LUSC, including natural killer cell mediated cytotoxicity, are immune-related pathways (Figure 4D). We did not perform a KEGG pathway analysis for LUAD or LUSC because fewer than one extrinsic pathway-related downregulated gene was found.

Next, we investigated how much the immune checkpoint inhibitory signal was activated in the group with a high expression of our intrinsic signature. The relationship between the intrinsic signature and other immune checkpoint targets was investigated in the NanoString and TCGA datasets (Figure 4E). In the TCGA LUAD dataset, the intrinsic signature correlated significantly and positively with CD274, LAG3, TIGIT, TIM3, PDCD1, and PDCD1LG2, with correlation coefficients ranging from 0.301 to 0.1. In the NanoString LUAD dataset, the intrinsic signature correlated positively with CD274, LAG3, and PDCD1LG2, and all the correlation coefficients were greater than 0.25. In the TCGA LUSC dataset, the intrinsic signature correlated significantly and positively with TIGIT, VISTA, TIM3, CTLA-4, PDCD1, and PDCD1LG2, and the correlation coefficients ranged from 0.11 to 0.340. In the NanoString LUSC dataset, the intrinsic signature was also positively correlated with CD274, LAG3, TIGIT, TIM3, CTLA-4, PDCD1, and PDCD1LG2, and all the correlation coefficients were greater than 0.25.

### 3.4. Differences in Transcriptional Patterns among NSCLC, LUAD, and LUSC

We derived the average AUC and accuracy of the six machine learning models to compare the predictive ability of each of them for NSCLC, LUAD, and LUSC (Figure 5A). The intrinsic gene signature of LUAD (AUC = 0.957 and accuracy = 0.9) had higher predictive power than the extrinsic or intrinsic gene signature of NSCLC overall (AUC = 0.792, accuracy = 0.81 and AUC = 0.805, accuracy = 0.816, respectively). The predictive power of the intrinsic gene signature of LUSC (AUC = 0.797 and accuracy = 0.833) was similar to that of the extrinsic and intrinsic gene signatures of NSCLC. Because only the intrinsic gene signature of LUAD has strong predictive power, we investigated differences in the genetic profiles of NSCLC, LUAD, and LUSC. Most of the top 100 genes in the ANOVA for LUAD were intrinsic genes (69%), whereas in NSCLC and LUSC, most genes were extrinsic (64% and 67%, respectively) (Figure 5B). These results indicate that LUAD and LUSC might be modulated by different PD-L1/PD-1 mechanisms. Because PD-L1/PD-1 expression is regulated by various pathways, we looked for an overlapping pathway in LUAD and LUSC. The number of genes overlapping with the overall NSCLC results was higher in LUSC than in LUAD (56 vs. 41, Figure 5C). In LUAD, the number of genes overlapping with the overall NSCLC results was similar in the intrinsic and extrinsic pathways (21 vs. 20). However, in LUSC, the number of genes overlapping with the overall NSCLC results in the extrinsic pathway was higher than that in the intrinsic pathway (44 vs. 12). We then looked for overlapping genes in LUAD and LUSC. Among the 100 total genes, 15 genes were found in both LUAD and LUSC. Only 2 of them (GREM1, AURKA) were intrinsic genes; the other 13 genes (LAG3, IL15RA, TAP1, NKG7, CCL2, CYBB, CD68, CD274, CD14, CCR1, FCGR3A/B, STAT1, and PDCD1LG2) were extrinsic.

## 4. Discussion

Our KEGG enrichment analysis revealed that the intrinsic genes in LUAD were significantly enriched in the cell cycle pathway. The intrinsic genes of both LUAD and LUSC were significantly enriched in the cellular senescence pathway. Previous studies have shown that PD-L1 expression is strongly influenced by cell cycle-related factors. CDK4/6 inhibition elevates PD-L1 protein levels in mouse embryonic fibroblasts, and the depletion of cyclin D1 also upregulated the PD-L1 protein level [20]. The expression of PD-L1 differed according to the cell cycle in an HNSCC cell line [21]. The cellular localization of PD-L1 also varies in specific cell cycle phases [21]. NEK2 is an important cell cycle regulator with functions such as centrosome separation and microtubule organization and stabilization. NEK2 is known to interact with PD-L1, and it inhibits the degradation of PD-L1 that is mediated by the ubiquitin-proteasome pathway. The simultaneous inhibition of NEK2 and PD-L1 was found to significantly inhibit the development of pancreatic cancer in a preclinical model [22]. Several studies have provided evidence that the cellular senescence pathway is involved in PD-L1/PD1. Amphiregulin produced from senescent stromal cells enhances PD-L1 expression in cancer cells and creates an immunosuppressive tumor microenvironment [23]. The inhibition of amphiregulin promotes tumor regression and suppresses chemical resistance in vivo [23]. The therapy-induced senescence-associated secretory phenotype (SASP) enhances the response to anti-PD-1 antibodies and improves survival in patients with ovarian tumors [24]. In melanoma, the generation of the inflammatory SASP by CDK4/6 inhibitors overcomes the immune checkpoint blockade resistance in a CD8+ T cell-dependent manner [25]. The SASP generated by the combination of MEK and a CDK4/6 inhibitor induces the accumulation of CD8+ T cells and makes pancreatic ductal adenocarcinoma more sensitive to immune checkpoint blockade [26].

Our intrinsic gene signature in LUAD has previously been reported to be related to the PD-L1/PD-1 pathway. MYC has recently been shown to transcriptionally induce PD-L1 expression by binding to the PD-L1 promoter [27]. The inhibition of RRM2 decreased the expression of PD-L1, and the tumor regression effect was better when both RRM2 and PD-1 were inhibited [28]. MYBL2 correlated positively with immune checkpoints, including TIGIT, SIGLEC15, PDCD1LG2, PDCD1, and LAG3 in LUAD [29]. AXL expression correlated significantly with the expression of PD-L1 and CXC chemokine receptor 6 genes in LUAD [30]. High CDCA5 expression correlated positively with the expression of PD-1 and PD-L1 in LUAD [31].

The genes with high ANOVA values in NSCLC overall were mostly extrinsic, but intrinsic genes were dominant in LUAD because the extrinsic genes in LUAD and LUSC overlap, but the intrinsic genes do not. Those overlapping extrinsic genes have a greater effect on NSCLC overall, and thus the extrinsic genes predominate.

Ayers’ extended immune gene signature correlated with prognosis in LUSC but not in LUAD. This result is also evidence that the intrinsic gene signature is more useful in LUAD, which is further supported by the fact that most genes with high ANOVA values in LUAD are intrinsic. Therefore, the PD-L1/PD-1 pathway is regulated by different pathways in LUAD and LUSC, and different gene signatures must be used to predict prognosis in the two conditions.

The intrinsic gene signature predicted a good prognosis for both cell types in our patient group; however, in the TCGA data, the group with a high intrinsic gene signature had a worse prognosis. Because our intrinsic gene signature is positively correlated with various immune checkpoints, we suggest that the immune inhibitory signal is activated in tumors with a high intrinsic gene signature. If immune checkpoint blockers are not administered in this patient group, the prognosis is expected to be poor due to dysfunction in anti-tumor immunity. The patient group with a high intrinsic gene signature in TCGA had a poor prognosis, but the effect of a PD-1 inhibitor is greater in that group. Therefore treatment with a PD-1 inhibitor is absolutely necessary in patients with a high intrinsic gene signature.

Among the intrinsic genes identified in the LUAD group, there is a gene for which target therapy is being developed. Therefore, a PD-1 inhibitor and a gene-targeted therapy could be a combination therapy. FDI-6 (forkhead domain inhibitor 6) inhibits the proliferation of LUAD cells and suppresses the activities of MYBL2 and FOXM1 [32]. R428, a selective AXL inhibitor, increases the survival rate of breast cancer patients by suppressing AXL [33]. Combined inhibition of PD-L1/PD-1 and TGF-β signaling is already under clinical trials in patients with metastatic colorectal or gastric cancer [34]. Inhibition of RRM2 enhanced the antitumor effect of PD-1 blockade in renal cancer by promoting CD8+ T cell infiltration [28].

Our study has several limitations. First, despite the relatively small sample size, our results were not validated by an external validation set. Most past biomarker studies on PD-1 inhibitors have focused on genomic mutations, and studies of transcriptional profiles are rare. Of those few transcriptional profile studies, most have focused on immune-related gene signatures, and almost no subgroup analyses have been conducted according to cell type. Therefore, external verification of our results is difficult. Second, we classified intrinsic and extrinsic pathways using pathway annotations provided by the NanoString company. However, because some genes can affect several different pathways, some genes we defined as intrinsic might also affect extrinsic pathways.

## 5. Conclusions

In summary, we found hitherto undiscovered intrinsic gene signatures in LUAD and LUSC that are well predictive of patient responsiveness to PD-1 inhibitors. The intrinsic gene signature can be particularly useful in LUAD, suggesting that the intrinsic pathway plays an important role in regulating the PD-L1/PD-1 pathway in LUAD. Patients with high intrinsic gene signatures should be treated with PD-1 inhibitors because the prognosis is poor if not treated with a PD-1 inhibitor.

## Figures and Tables

**Figure 1 cancers-15-00197-f001:**
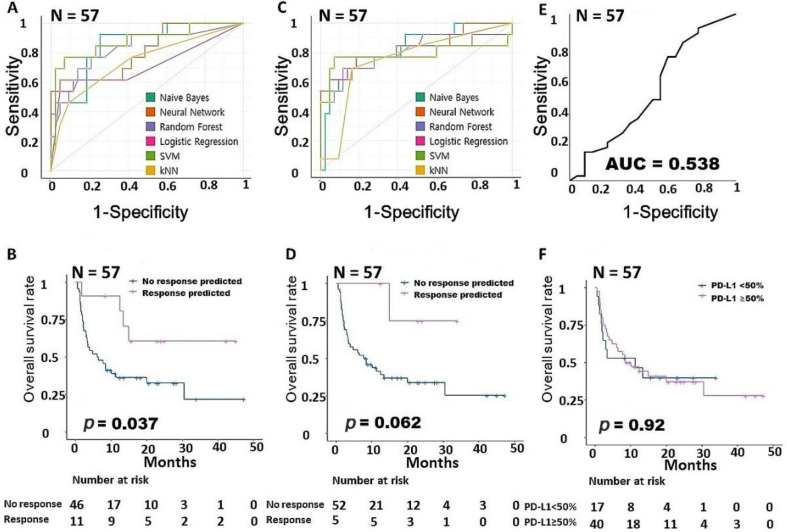
Building a predictive model for responsiveness to PD-1 inhibitors in NSCLC. (**A**) Comparison of the AUCs for the intrinsic gene signature according to six machine learning models. (**B**) Survival analysis according to the intrinsic gene signature. (**C**) Comparison of the AUCs for the extrinsic gene signature according to six machine learning models. (**D**) Survival analysis according to the extrinsic gene signature. (**E**) AUC according to PD-L1 expression. (**F**) Survival analysis according to PD-L1 expression. AUC, area under the ROC curve; NSCLC, non–small cell lung cancer; PD-1, programmed cell death protein 1; PD-L1, programmed cell death-ligand 1.

**Figure 2 cancers-15-00197-f002:**
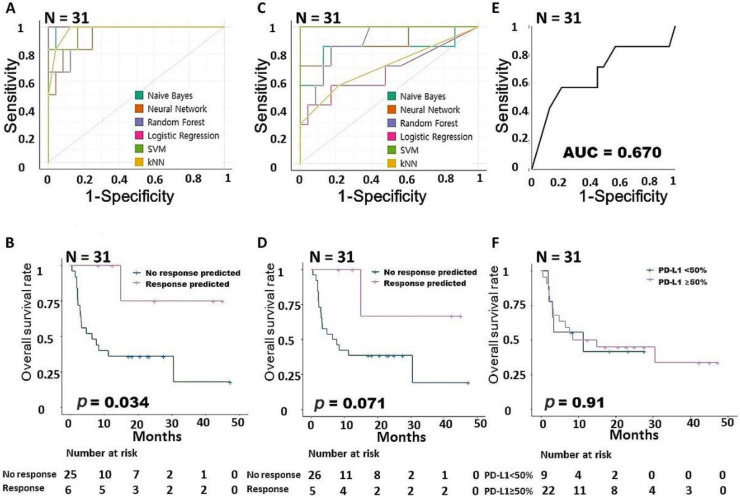
Building a predictive model for responsiveness to PD-1 inhibitors in LUAD. (**A**) Comparison of the AUCs for the intrinsic gene signature according to six machine learning models. (**B**) Survival analysis according to the intrinsic gene signature. (**C**) Comparison of the AUCs for the extrinsic gene signature according to six machine learning models. (**D**) Survival analysis according to the extrinsic gene signature. (**E**) AUC according to PD-L1 expression. (**F**) Survival analysis according to PD-L1 expression. AUC, area under the ROC curve; LUAD, lung adenocarcinoma; PD-1, programmed cell death protein 1; PD-L1, programmed cell death-ligand 1.

**Figure 3 cancers-15-00197-f003:**
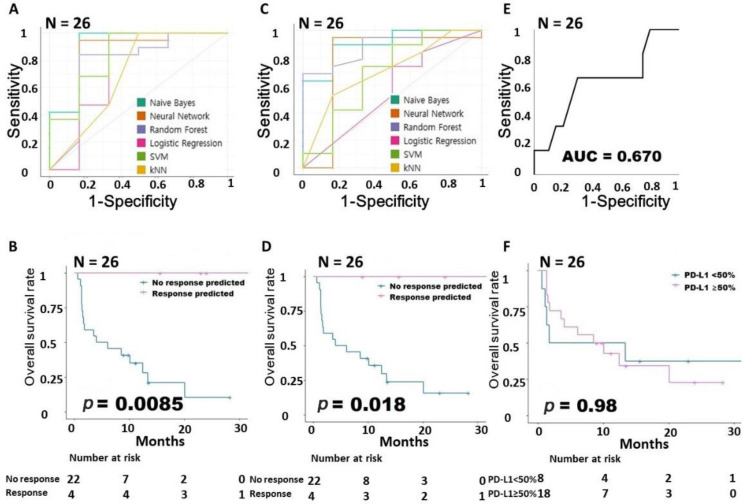
Building a predictive model for responsiveness to PD-1 inhibitors in LUSC. (**A**) Comparison of the AUCs for the intrinsic gene signature according to six machine learning models. (**B**) Survival analysis according to the intrinsic gene signature. (**C**) Comparison of the AUCs for the extrinsic gene signature according to six machine learning models. (**D**) Survival analysis according to the extrinsic gene signature. (**E**) AUC according to PD-L1 expression. (**F**) Survival analysis according to PD-L1 expression. AUC, area under the ROC curve; LUSC, lung squamous cell carcinoma; PD-1, programmed cell death protein 1; PD-L1, programmed cell death-ligand 1.

**Figure 4 cancers-15-00197-f004:**
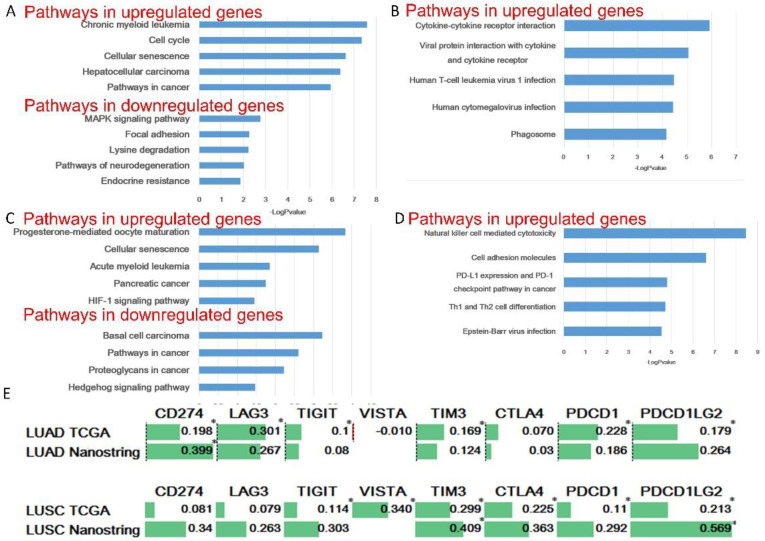
Pathway enrichment analysis of genes associated with responsiveness to PD-1 inhibitor treatment. (**A**) KEGG pathway terms related to intrinsic genes in LUAD. (**B**) KEGG pathway terms related to extrinsic genes in LUAD. (**C**) KEGG pathway terms related to intrinsic genes in LUSC. (**D**) KEGG pathway terms related to extrinsic genes in LUSC. (**E**) Correlation analysis between the intrinsic gene signature and other immune checkpoint targets. LUAD, lung adenocarcinoma; LUSC, lung squamous cell carcinoma; PD-1, programmed cell death protein 1; TCGA, The Cancer Genome Atlas. * Spearman’s rank correlation coefficient.

**Figure 5 cancers-15-00197-f005:**
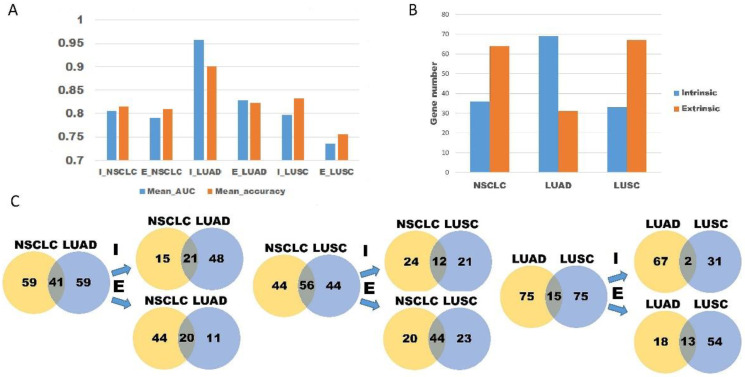
Differences in transcriptional patterns among NSCLC, LUAD, and LUSC. (**A**) The mean AUC and accuracy of six machine learning models according to the cellular subtype and intrinsic or extrinsic pathway. (**B**) Differences in the proportion of intrinsic and extrinsic genes among NSCLC, LUAD, and LUSC. (**C**) Overlapping patterns of intrinsic and extrinsic genes in NSCLC, LUAD, and LUSC. AUC, area under the ROC curve; E, extrinsic; I, intrinsic; LUAD, lung adenocarcinoma; LUSC, lung squamous cell carcinoma; NSCLC, non-small cell lung cancer.

**Table 1 cancers-15-00197-t001:** The gene list of each pathway.

**30 intrinsic pathway related genes in a total NSCLC**
GREM1, AURKA, AXL, RRM2, CDC25A, PRIM1, CCNB1, ATP7A, PCLAF, LMNB1, ERBB2, NSD1, KAT6B, CDCA8, KIF2C, SRM, GPX1, PIK3CB, MYBL2, RUVBL1, UBE2C, RBBP5, FASLG, ARID4B, NCAPG, PIK3R5, CAV1, BAX, MYC, ITGA5.
**5 extrinsic pathway related genes in a total NSCLC**
IL15RA, CCR1, CCL2, CYBB, FCER1G
**15 intrinsic pathway related genes in lung adenocarcinoma**
MYC, RRM2, MAPK14, MYBL2, AXL, CDCA5, AKT1S1, TGFB1, CDK12, PCLAF, BAX, HMOX1, ARID4A, SLC1A5, ARID4B.
**15 extrinsic pathway related genes in lung adenocarcinoma**
CD209, CCL13, LAG3, BCL2L1, HLA-C, IL15RA, MSLN, TAP1, CD8A, NKG7, TAP2, MST1R, CCL5, CCL2, CYBB.
**5 intrinsic pathway related genes in lung squamous cell carcinoma**
SERPINE1, PIK3R5, PIK3CB, RPA3, KIF2C
**10 extrinsic pathway related genes in lung squamous cell carcinoma**
CCR1, FCER1G, CD38, GNLY, CYBB, IL15RA, CCL2, VAV1, CD274, GZMB

## Data Availability

Our gene expression and response data were included in the Appendix A.

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
