# Peer review of "Intrinsic and Extrinsic Transcriptional Profiles That Affect the Clinical Response to PD-1 Inhibitors in Patients with Non–Small Cell Lung Cancer"

_cancers, 2022, doi:10.3390/cancers15010197_

Round 1
Reviewer 1 Report
The authors investigated here the use of a machine learning method elaborating intrinsic and extrinsic transcriptional profiles possibly involved in clinical response to PD-1 inhibitors in a cohort of 57 NSCLCs.
Intrinsic genes in lung adenocarcinoma showed higher levels than what observed in NSCLC overall or lung squamous cell carcinoma, then better predicting response to PD-1 inhibitors.
Of note, cell cycle and cellular senescence pathways were related to the upregulation of intrinsic genes in adenocarcinoma and a direct correlation between intrinsci genes in adenocarcinoma and other immune checkpoint targets, such as CD274, LAG3, and PDCD1LG2.
The study is interesting and I have just minor comments, as follows:
1. Please spell out the type of specimens used for mRNA anlaysis: cytology ? biopsies ? surgical samples ?
2. Please spell out PD-L1 immunohistochemical details on the type of mAB clone and automated immunostainer used in the study.
3. This is a complex molecular investigation that cannot be likely replicated on routine practice. Is there some messages from the study herein that can be translated in real life ?
Author Response
- Please spell out the type of specimens used for mRNA anlaysis: cytology ? biopsies ? surgical samples ?
Response: We added the type of specimens used for mRNA analysis in supplementary Table 1. Biopsy was performed in 40 cases and surgical resection was performed in 17 cases. Inoperable cases were treated with PD-1 inhibitors, and surgical cases were treated with PD-1 inhibitors for subsequent recurrence or metastasis. We added this comment in materials and methods section
- Please spell out PD-L1 immunohistochemical details on the type of mAB clone and automated immunostainer used in the study.
Response: We added PD-L1 immunohistochemical details on the type of mAB clone and automated immunostainer used in the study in materials and methods section. “The PD-L1 immunohistochemistry (clone: SP263, rabbit monoclonal, Roche, Basel, Switzerland) was performed using the OptiView DAB Immunohistochemical Detection Kit on a Ventana BenchMark ULTRA instrument.”
- This is a complex molecular investigation that cannot be likely replicated on routine practice. Is there some messages from the study herein that can be translated in real life?
Response: Among the intrinsic genes identified in the LUAD group, there is a gene for which target therapy is being developed. Therefore, a PD-1 inhibitor and a gene-targeted therapy could be a combination therapy. FDI-6 (forkhead domain inhibitor 6) inhibits the prolifer-ation of LUAD cells and suppresses the activities of MYBL2 and FOXM1[32]. R428, a se-lective AXL inhibitor, increases the survival rate of breast cancer patients by suppressing AXL[33]. Combined inhibition of PD-L1/PD-1 and TGF-β signaling is already under clin-ical trials in patients with metastatic colorectal or gastric cancer [34]. Inhibition of RRM2 enhanced the antitumor effect of PD-1 blockade in renal cancer by promoting CD8+ T cell infiltration[28]. We added these comments in discussion section
Reviewer 2 Report
1. The authors' research has certain guiding significance, through the RNA sequencing analysis of the samples they collected, the model is constructed with a high AUC value;
2. Make the following comments on the study:
(1). Overall, the sample is single-center and small, and there are only a few data in some subgroups, so the evidence level is insufficient;
(2). 40% of patients in the lung adenocarcinoma subgroup carry EGFR driver mutations, and this type of factors that are likely to affect the efficacy of immunity is not included in the analysis;
(3). The study model lacks an exact match of external validation.
Author Response
(1). Overall, the sample is single-center and small, and there are only a few data in some subgroups, so the evidence level is insufficient;
Response: We also agree that sample of our study is small, and there are only a few data in some subgroups. It is ideal to verify on more samples. However, since most patients receiving PD-1 inhibitor treatment only undergo biopsy, very few specimens are available for gene expression testing. Because sufficient RNA is essential for nanostring gene expression analysis, there were not many samples suitable for the experiment. However, we studied the intrinsic gene signature, which has received little attention. Also, because LUAD and LUSC have different mechanisms of development and different treatments, subgroup analysis was essential. We obtained significantly different results in intrinsic gene expression analysis between LUAD and LUSC. Our results are a small sample and will have to be validated in other researchers' experiments.
(2). 40% of patients in the lung adenocarcinoma subgroup carry EGFR driver mutations, and this type of factors that are likely to affect the efficacy of immunity is not included in the analysis;
Response: We also agree that EGFR mutations can affect the response of PD-1 inhibitors. Therefore, we performed a subgroup analysis according to EGFR mutation. Since all cases with EGFR mutation were non-response, only the group without EGFR mutation was analyzed. The AUC values for the 15 intrinsic pathway–associated genes ranged from 0.87 to 1 (NB: 1, NN: 1, RF: 0.870, LR: 0.987, SVM: 0.987, kNN: 0.883, sum of AUCs: 5.727, supplementary Figure 2A). The AUC values for the 15 extrinsic pathway–associated genes ranged from 0.5 to 1 (NB: 0.922 NN: 0.961, RF: 0.805, LR: 0.506, SVM: 1, kNN: 0.786, sum of AUCs: 4.98, supplementary Figure 2B). Therefore, even in the EGFR-negative group, intrinsic pathway-associated genes showed slightly higher predictive power than extrinsic. We added this result in results section.
(3). The study model lacks an exact match of external validation.
Response: We also agree that our study model lacks an exact match of external validation. Most past biomarker studies of PD-1 inhibitors have focused on genomic mutations, and studies on their transcriptional profiles have been rare. Intrinsic gene signatures have recently attracted attention, and in the past, only extrinsic gene signatures (interferon gamma signature, J Clin Invest. 2017 Aug 1;127(8):2930-2940, T cell–inflamed signature, Cancer Immunol Res. 2018 Sep;6(9):990-1000) have been studied. Therefore, please consider the lack of past datasets that can be validated. Although it was not verified in the group using the PD-1 inhibitor, it was verified in the TCGA dataset without using the PD-1 inhibitor. In the TCGA dataset, intrinsic gene signatures were correlated with poor prognosis. Furthermore. our intrinsic gene signature is positively correlated with various immune checkpoints, a group with a high intrinsic gene signature is predicted to respond better to PD-1 inhibitors.
Reviewer 3 Report
This is an interesting and novel study showing intrinsic and extrinsic transcriptional profiles that affect the clinical response to PD-1 inhibitors in NSCLC patients. Specifically, authors find that the intrinsic gene signature was a good predictive or prognostic biomarker in lung adenocarcinoma (LUAD) patients. The manuscript is well written and discussed. I only have several minor changes to be addressed prior to be considered for publication in Cancers:
1) Authors should include statistical analyses performed, as well as the number of patients used in each figure.
2) The quality of figures should be improved
Author Response
1) Authors should include statistical analyses performed, as well as the number of patients used in each figure.
Response: We added the number of patients used in each figure.
2) The quality of figures should be improved
Response: We improved the quality of figures by increasing the size of numbers or letters.
Round 2
Reviewer 2 Report
As the authors say, their research, which has not received much attention so far, is an interesting and even potentially very meaningful piece of work. In the meantime, I hope the authors will continue to refine this work, either by expanding the sample size or by querying external validation, to obtain even better results.